# DLM: A Scalable Decision Language Model for Multi-Agent Sequential Decision Across Tasks

## Abstract

Building a scalable model from offline datasets to tackle a broad spectrum of multi-agent sequential decision-making problems across tasks is a crucial step toward reusable and generalizable decision intelligence. However, the mainstream offline multi-agent reinforcement learning (MARL) methods lack generalization due to their reliance on fixed observation formats and action spaces. In contrast, language models offer flexible input representations that are not constrained by predefined dimensions. Motivated by this, we propose the decision language model (DLM), a framework that formulates decision-making as a dialogue-style sequence prediction problem. DLM is trained in two stages: a supervised fine-tuning (SFT) phase that leverages dialogue-style datasets to enable centralized training with inter-agent context, generating coordinated actions consistent with environment constraints; and a group relative policy optimization (GRPO) phase that further trains DLM-SFT to enhance robustness to out-of-distribution (OOD) actions through lightweight reward functions, yielding DLM-GRPO. Despite its simple design, DLM-SFT matches the performance of leading offline MARL methods across all tasks on the benchmark using only observation and action data. DLM-GRPO further improves execution reliability by significantly reducing OOD action risks and achieves strong zero-shot generalization to unseen tasks, reaching state-of-the-art performance with a single unified model.

## 1 Introduction

Large language models (LLMs) (Touvron et al., 2023; Bai et al., 2023; Ouyang et al., 2022; Brown et al., 2020), trained on diverse offline datasets, have demonstrated remarkable generalization in a wide range of downstream tasks in natural language processing (Guo et al., 2024; Schick et al., 2023). Nevertheless, LLMs often fall short when applied to sequential decision-making problems due to misalignment with task goals and environment dynamics (Ahn et al., 2022), which are better handled by reinforcement learning (RL). In contrast, online RL (Sutton & Barto, 1998; Zhang et al., 2025b; Schulman et al., 2017; Haarnoja et al., 2018) relies on repeated environment interaction, which is often expensive, inefficient, and risky in real-world applications. As a result, offline RL (Kumar et al., 2020; Fujimoto et al., 2019; Kostrikov et al., 2021) has emerged as a promising alternative that learns from static datasets without additional interaction. However, most existing offline RL efforts focus on improving performance within a given task by mitigating out-of-distribution (OOD) issues (Kumar et al., 2019; Zhang et al., 2025a; Levine et al., 2020) and optimizing from limited data coverage. Although recent offline RL methods advance policy learning from fixed datasets, they typically focus on single-task performance and overlook the challenge of generalization across diverse tasks. At the same time, real-world applications such as autonomous driving (Zhou et al., 2024), collaborative robotics (Seraj et al., 2023), and strategic games (Rashid et al., 2020; Liu et al., 2025) demand both generalization across various tasks and scalability with increasing numbers of interacting agents. Addressing these challenges requires a unified model capable of handling multi-agent sequential decision across tasks within a single framework.

A key reason for the limited generalization in RL lies in the rigid construction of models. States and actions are encoded in fixed formats that are tightly coupled with specific task definitions, hindering transferability between environments with different input and output structures. Recent efforts

such as the decision transformer (DT) (Chen et al., 2021) and the trajectory transformer (TT) (Janner et al., 2021) address this limitation in single agent settings by casting decision-making as sequence modeling, allowing a more flexible and data-driven formulation. However, extending these methods to multi-task scenarios introduces a representation mismatch challenge. Most existing approaches perform modality-specific embeddings by normalizing and discretizing continuous inputs into bounded index representations. Although this encoding strategy works in single-task settings, it struggles to generalize across tasks due to the large variability in input distributions, making the representation unstable and task dependent. The situation becomes even more complex in multi-agent settings, which introduce additional coordination and scalability challenges. One key challenge is incompatibility of centralized training with decentralized execution (CTDE) (Lowe et al., 2017). While the CTDE paradigm aims to leverage global information during training and rely on local observations at execution, reconciling these two modes within a single model remains difficult, as it requires balancing global context with agent-level autonomy. Another challenge is the scarcity of individual rewards. In multi-agent environments, rewards are usually provided at the global level rather than for each agent individually, making it difficult to apply sequence modeling approaches that depend on each agent's reward and introducing the credit assignment problem. Furthermore, unlike observations and actions that are readily available in offline datasets, step-level reward signals are often difficult to obtain, further limiting the use of reward-conditioned models.

In this paper, we propose the decision language model (DLM), a scalable framework that addresses the three key challenges of multi-agent sequential decision-making across tasks while also reducing OOD errors under limited data. DLM is trained in two stages: a supervised fine-tuning (SFT) phase, which aligns a pre-trained language model with the decision domain, referred to as DLM-SFT; and a group relative policy optimization (GRPO) (Shao et al., 2024) phase that enhances robustness to OOD actions, referred to as DLM-GRPO. To tackle the representation mismatch, we convert observations and actions into natural language and reformulate decision-making as dialogue-style sequence modeling, allowing language models to flexibly encode diverse tasks through tokenization. To address the CTDE incompatibility, we design a dialogue-style trajectory representation for each agent, which preserves inter-agent context during centralized training while supporting decentralized execution from local observations. To handle the individual reward scarcity, we discard reward data and avoid training from scratch, instead fine-tuning a pre-trained language model capable of implicit credit assignment via temporal attention. Finally, we mitigate OOD risks by introducing a simple executability-based reward during GRPO to penalize invalid actions.

We extensively evaluate DLM on multiple tasks from the StarCraft Multi-Agent Challenge (SMAC) (Samvelyan et al., 2019) and SMACv2 (Ellis et al., 2023). The results show that DLM-SFT, despite not using reward data, performs competitively with the leading offline MARL methods. DLM-GRPO further improves performance by reducing OOD actions, achieving state-of-the-art (SOTA) results and demonstrating strong zero-shot generalization to unseen tasks.Overall, our study demonstrates the effectiveness of DLM in handling diverse multi-agent decision tasks. The following is a list of the contributions of this paper.

- We propose DLM, a framework for multi-agent sequential decision across tasks, trained with a two-stage method improving OOD robustness, addressing representation mismatch, CTDE incompatibility, and reward scarcity without relying on online interaction.
- We introduce a dialogue-style offline dataset construction method for MARL and design an LLM-based post-training approach aligned with the CTDE paradigm, enabling centralized training with inter-agent context and decentralized execution from local observations.
- We demonstrate that DLM achieves SOTA performance across multiple tasks in the SMAC benchmark using a single model, exhibits strong zero-shot generalization to unseen tasks, and may provide a pathway toward constructing general decision-making models.

## 2 RELATED WORKS

**Offline MARL** Offline RL aims to learn policies from fixed datasets without further environment interaction, making it suitable for high-risk or cost-sensitive domains. A common approach is to apply behavior cloning (BC) (Syed et al., 2008), which directly imitates actions in the dataset. While simple and stable, BC does not account for the distributional shift between the training data and the learned policy's behavior, often resulting in compounding errors during deployment. To address

this, methods such as TD3+BC (Kostrikov et al., 2021) and CQL (Kumar et al., 2020) introduce value-based regularization to penalize unseen or high-risk actions and reduce overestimation bias, achieving better performance in the single-agent setting. In the multi-agent setting, these issues become even more severe due to the exponential growth of the joint state-action space and the need for coordinated behavior. MACQL (Formanek et al., 2024) extends CQL to the multi-agent regime by applying conservative value estimation under CTDE paradigm. OMIGA (Wang et al., 2024) improves coordination by shaping local policies with implicit global-to-local value information. CFCQL (Shao et al., 2023) further enhances robustness through agent-level counterfactual regularization, enabling stable learning even under partial observability or suboptimal data coverage. Despite these advances, existing offline MARL methods are tied to task-specific architectures or value-centric learning objectives, limiting their scalability across diverse agents and environments.

**Sequence Modeling for Decision Making**    Recent advances have recast RL as a sequence modeling problem, enabling the use of large-scale Transformer (Vaswani et al., 2017) architectures originally developed for language understanding. The DT (Chen et al., 2021) pioneers this perspective by predicting actions autoregressively, conditioned on the return-to-go, past states, and actions. This formulation bypasses the need for bootstrapped value estimation and instead treats policy learning as a supervised sequence prediction task. Building on this, the TT (Janner et al., 2021) extends this idea by modeling full trajectory distributions with tokenized inputs, improving sample efficiency. Pushing further toward generalist agents, Gato (Reed et al., 2022) unifies vision, language, and control tasks through sequence modeling, showing that a single transformer can operate across diverse domains. In the multi-agent setting, however, sequence modeling remains limited. Multi-agent decision transformer (MADT) (Meng et al., 2023) shares parameters across agents to enable independent pre-training, and applies online fine-tuning with a centralized critic. While this design simplifies training, it neglects inter-agent coordination during pre-training and relies on environment interaction during fine-tuning, deviating from the offline paradigm.

**Aligning Pre-trained LLMs for Decision-Making**    Pre-trained LLMs provide strong priors that help agents make informed decisions with minimal exploration, making them attractive for offline decision-making. However, these priors are often misaligned with target tasks or environments, motivating adaptation through alignment techniques. SFT, together with parameter-efficient methods such as LoRA (Hu et al., 2021), enables scalable adaptation in multi-agent settings. Beyond SFT, reinforcement learning from human feedback (RLHF) (Ouyang et al., 2022) further refines model behavior using preference-based rewards, yielding more robust policies. More recently, GRPO (Shao et al., 2024) has simplified alignment by leveraging lightweight heuristic rewards, making it particularly suitable for offline RL where explicit reward design is costly or infeasible.

## 3 METHOD

In this section, we present the DLM, a scalable framework for multi-agent sequential decision-making cross tasks. As illustrated in Fig. 1, DLM training follows a four-step pipeline: two data preparation stages and two model training stages. Due to the lack of suitable offline datasets covering all tasks, we first construct a comprehensive offline multi-task dataset and transform the collected trajectories into a dialogue-style sequence representation. The dataset is then partitioned into two subsets for SFT and GRPO, respectively. We initialize DLM with the pre-trained LLaMA-3.2-1B model (Grattafiori et al., 2024) and fine-tune it through SFT to obtain the DLM-SFT model, enabling the model to generate valid and rule-compliant actions. Next, we identify the OOD-prone samples where DLM-SFT exhibits suboptimal performance and apply GRPO with simple executability-based rewards to further refine the policy and enhance its robustness under distributional shifts.

**Problem Formulation**    We model the cooperative multi-agent sequential decision-making problem as a decentralized partially observable markov decision process (Dec-POMDP) (Oliehoek & Amato, 2016), defined by the tuple $\mathcal{G} = \langle \mathcal{N}, \mathcal{S}, \mathcal{A}, P, \Omega, O, R, \gamma \rangle$. Here, $\mathcal{N} = \{1, \ldots, n\}$ denotes the set of agents, $\mathcal{S}$ is the set of global states, and $\mathcal{A} = \prod_{i=1}^{n} \mathcal{A}^i$ is the joint action space, where $\mathcal{A}^i$ is the action space for agent $i$. At each time step $t$, the environment is in state $s_t \in \mathcal{S}$, and each agent $i$ receives a private observation $o_t^i \in \Omega^i$, where $\Omega = \prod_{i=1}^{n} \Omega^i$ is the joint observation space and $O : \mathcal{S} \to \Delta(\Omega)$ is the observation function. Based on its local observation, each agent selects an

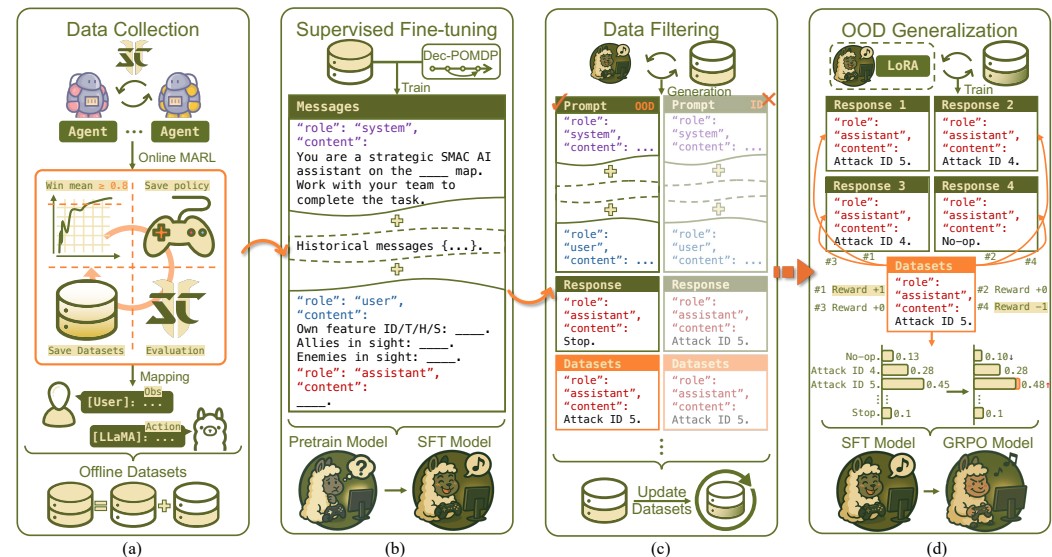

Figure 1: Overall training pipeline of DLM. (a) Offline data collected from online MARL algorithms is transformed into dialogue-style sequences and split into two subsets. (b) The pre-trained model is fine-tuned on the first subset via SFT to align with the decision domain, resulting in DLM-SFT. (c) DLM-SFT generates policies on the second subset, and OOD-prone samples are filtered by comparing outputs with the dataset. (d) GRPO further trains DLM-SFT on the filtered subset using lightweight executability-based rewards, yielding the final DLM-GRPO model.

action according to its individual policy $\pi^i : \Omega^i \to \Delta(\mathcal{A}^i)$. The joint action $\boldsymbol{a}_t = (a_t^1, \ldots, a_t^n)$ induces a transition to the next state $s_{t+1}$ according to the environment dynamics $P : \mathcal{S} \times \mathcal{A} \to \Delta(\mathcal{S})$. The system receives a global reward $r_t = R(s_t, \boldsymbol{a}_t)$, and $\gamma \in (0, 1)$ denotes the discount factor governing future returns. In the offline setting, we assume access to a fixed dataset $\mathcal{D} = \{\tau^{(k)}\}_{k=1}^M$ consisting of $M$ collected trajectories. Each trajectory $\tau^{(k)}$ contains the key elements defined in the Dec-POMDP tuple $\mathcal{G}$. The goal is to learn decentralized policies $\{\pi^i\}_{i=1}^n$ to achieve cooperative behavior across diverse multi-agent tasks.

## 3.1 DIALOGUE-STYLE OFFLINE DATASET CONSTRUCTION

To address the representation mismatch challenge discussed in Sec. 1, we rethink the encoding of observations and actions in multi-task settings. The wide variability in their numerical ranges and structures across tasks makes fixed-format mappings impractical. In contrast, LLMs leverage tokenization techniques (Mikolov et al., 2013) to embed diverse concepts into a shared semantic space, enabling flexible representation learning.

Inspired by this, we verbalize multi-agent decision trajectories as natural language dialogues, framing sequential decision-making as a language modeling problem. Any decision process can be described by specifying the environment, the agent's observation, and the intended action in natural language. Specifically, taking

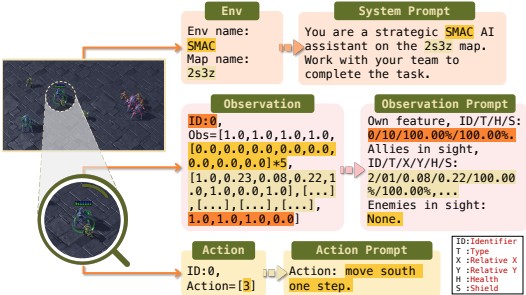

Figure 2: Mapping SMAC environment, observations, and actions into a dialogue-style prompt, with highlighted text showing key correspondences.

SMAC as an example, observations consist of four feature groups: move_feats, enemy_feats, ally_feats, and feats, encoding attributes such as position, health, and visibility. These features, originally designed for computational processing, can be naturally verbalized into textual

Figure 3: Training and inference frameworks for DLM. (a) Centralized training using dialogue-style trajectories with inter-agent information. (b) Decentralized inference where each agent independently generates actions based on its local trajectory.

descriptions. As illustrated in Fig. 2, environment information, agent observations, and actions are mapped into a fixed prompt format, where the highlighted text shows the correspondence between original features and their verbalized forms. As shown in Fig. 1(a), observations are treated as the `user` input and actions as the `assistant` reply, forming a dialogue turn. We further convert the dialogue into the LLaMA-3 chat format for compatibility, as detailed in Fig. 7. As existing datasets lack full coverage of SMAC tasks, we collect offline data following the dataset construction methodology used in D4RL (Fu et al., 2020). Specifically, we train TGCNet (Zhang et al., 2025b) until the win rate exceeds 80%, after which we save the model and use it to interact with the environment to generate trajectories. Details on dataset construction are provided in Appendix A.1.

## 3.2 SFT FOR MULTI-AGENT SEQUENTIAL DECISION

We adopt an autoregressive sequence modeling approach without relying on explicit value functions or rewards. Unlike prior single-agent formulations, we design a dialogue-style sequence representation that supports CTDE. Instead of pre-training from scratch, we initialize DLM with pre-trained language model and adapt it to the decision domain via SFT on half of the constructed offline dataset $\mathcal{D}_{\text{SFT}}$. The overall training and inference framework is presented in Fig. 3 and described in Alg. 1. The mathematical analysis and motivation behind this design are provided in the Appendix A.3.

**Dialogue-Style Sequence Representation** We represent multi-agent decision trajectories as structured sequences of observation-action pairs. Formally, each trajectory $\tau^{(k)}$ is defined as:

$$\tau^{(k)} = \left( \left\{ (o_t^{(k),i}, \, a_t^{(k),i}) \mid t = 1, \ldots, T^{(k)} \right\} \right)_{i=1}^{N}, \tag{1}$$

where $N$ is the number of agents and $T^{(k)}$ denotes the length of the $k$-th trajectory. Here, $o_t^{(k),i}$ represents the observation of agent $i$ at time $t$ within the $k$-th trajectory, and $a_t^{(k),i}$ denotes the corresponding action. Since each observation-action pair has already been verbalized into a dialogue-style format, the model inputs are constructed by stacking them in the order specified in Eq. 1. To improve scalability for the number of agents, we apply a maximum token limit, truncating sequences that exceed it and adopting dynamic packing strategies.

**SFT Training** We fine-tune DLM on the offline dataset, aligning the model's outputs with multi-agent decision demonstrations. In practice, DLM is trained to predict the next assistant reply (i.e., the action) conditioned on the dialogue history up to the current observation. Formally, the SFT objective minimizes the following loss:

$$\mathcal{L}_{\text{SFT}} = -\frac{1}{M} \sum_{k=1}^{M} \sum_{t=1}^{T^{(k)}} \sum_{i=1}^{N} \log P_\theta \left( a_t^{(k),i} \mid \tau_{\leq (o_t^{(k),i})}^{(k)} \right), \tag{2}$$

---

**Algorithm 1** SFT Training (Left) and Inference (Right) Procedures for DLM

1: **Input:** Offline dataset $\mathcal{D}_{\text{SFT}}$, pre-trained model parameters $\theta_{\text{init}}$.
2: **Initialize:** $\theta_{\text{SFT}} \leftarrow \theta_{\text{init}}$; tokenizer; system prompts; max token length $L$.
3: **for** each trajectory $\tau^{(k)} \in \mathcal{D}$ **do**
4:   Tokenize $\tau^{(k)}$ by stacking $(o_t^{(k),i}, a_t^{(k),i})$ pairs with system prompts, truncated to $L$.
5: **end for**
6: Construct mini-batches with dynamic packing.
7: **for** each mini-batch **do**
8:   Predict next action conditioned on dialogue history.
9:   Update $\theta_{\text{SFT}}$ by minimizing the loss in Eq. 2.
10: **end for**
11: **Output:** Fine-tuned model $\theta_{\text{SFT}}$.

1: **Input:** Fine-tuned model $\theta_{\text{SFT}}$, tokenizer, environment.
2: **for** each episode **do**
3:   Reset environment and initialize history.
4:   **while** episode not terminated **do**
5:     Encode observations into prompts.
6:     **for** each agent $i = 1, \ldots, N$ **do**
7:       Predict action according to Eq. 3.
8:       **if** action invalid or unavailable **then**
9:         Resample from Eq. 4.
10:       **end if**
11:     **end for**
12:     Execute actions and update history.
13:   **end while**
14:   Record episode win or loss outcome.
15: **end for**
16: **Output:** Test win rates.

---

where $M$ is the number of trajectories, $T^{(k)}$ is the length of the $k$-th trajectory, $N$ is the number of agents, and $\tau_{\leq (o_t^{(k),i})}^{(k)}$ denotes the dialogue history up to and including the current observation $o_t^{(k),i}$.

**Decentralized Inference**   During inference, DLM enables decentralized decision-making while maintaining the benefits of centralized training. Each agent independently generates its action based on its dialogue history, without requiring access to other agents' observations or actions. Formally, for each agent $i$ at time step $t$, the model predicts the next action by:

$$a_t^i = \arg\max_a P_{\theta_{\text{SFT}}}(a \mid \tau_{\leq (o_t^i)}). \tag{3}$$

If the predicted action is invalid or not allowed by the environment's available actions, we resample by drawing from the predicted distribution, sampling an action from the truncated distribution where only the top-$k$ tokens whose cumulative probability exceeds the top-$p$ threshold are considered:

$$a_t^i \sim P_{\theta_{\text{SFT}}}^{\text{top-}p,\text{top-}k}\left( \cdot \mid \tau_{\leq (o_t^i)} \right). \tag{4}$$

### 3.3 FILTERING OOD SAMPLES

Although DLM-SFT learns reasonable decision behaviors from offline data, it occasionally generates invalid or OOD actions due to the inherent limitations of dataset coverage. In multi-agent settings like SMAC, the observation-action space is vast and continuous, making exhaustive offline coverage impractical. As illustrated by the t-SNE visualization in Fig. 4, even with diverse trajectory collection across multiple tasks, the sampled observations and actions still only occupy a sparse subset of the overall space. This reveals a fundamental limitation: simply enlarging the dataset cannot completely eliminate OOD issues because the environment dynamics are effectively unbounded.

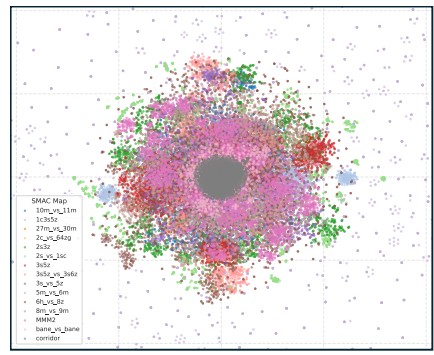

Figure 4:  t-SNE projection of observation distributions from the offline dataset across all SMAC tasks.

To address this, we apply OOD filtering on the other half of the offline dataset $\mathcal{D}_{\text{GRPO}}$. Specifically, for each observation $o^i$, we retain samples where the model-predicted action $\pi_{\theta_{\text{SFT}}}^i(o^i)$ either differs from the corresponding dataset action or violates the environment's executable action constraints. Formally, the filtered dataset $\mathcal{D}_{\text{OOD}}$ is defined as:

$$\mathcal{D}_{\text{OOD}} = \left\{ (o^i, a^i) \in \mathcal{D}_{\text{GRPO}} \mid \pi_{\theta_{\text{SFT}}}^i(o^i) \neq a^i \ \text{ or } \ \pi_{\theta_{\text{SFT}}}^i(o^i) \notin \mathcal{A}_{\text{avail}}(o^i) \right\}, \tag{5}$$

where $\mathcal{A}_{\text{avail}}(o^i)$ is the set of available actions based on agent $i$'s local observation $o^i$. The filtering procedure is illustrated in Fig. 1(c). This filtering strategy selectively retains challenging or misaligned samples, allowing subsequent GRPO training to focus on improving robustness against OOD behaviors while reducing overall training cost.

### 3.4 PREFERENCE OPTIMIZATION FOR OOD GENERALIZATION

To further improve action feasibility and policy alignment, we introduce a preference optimization stage that penalizes OOD behaviors and encourages consistency with the filtered $\mathcal{D}_{\text{OOD}}$ dataset, as shown in Fig. 1(d). Unlike RLHF-style approaches that rely on learning an additional reward model, we adopt simple handcrafted objectives to avoid the challenges of reward estimation in offline multi-agent sequential decision tasks. Specifically, we optimize two criteria: (1) ensuring executability under environment constraints while maintaining generalization, and (2) promoting agreement with actions from $\mathcal{D}_{\text{OOD}}$. We define a lightweight preference reward $r(o^i, a^i)$ based on these two criteria:

$$
r(o^i, a^i) = \begin{cases} 1, & \text{if } a^i = \hat{a}^i \text{ and } a^i \in \mathcal{A}_{\text{avail}}(o^i), \\ 0, & \text{if } a^i \neq \hat{a}^i \text{ and } a^i \in \mathcal{A}_{\text{avail}}(o^i), \\ -1, & \text{if } a^i \notin \mathcal{A}_{\text{avail}}(o^i), \end{cases} \tag{6}
$$

where $\hat{a}^i$ denotes the dataset action paired with $o^i$ in $\mathcal{D}_{\text{OOD}}$. Given $\mathcal{D}_{\text{OOD}}$ and the corresponding preference rewards, we perform GRPO on the LoRA-adapted DLM-SFT model to refine its decision-making alignment. For each $o^i$, we sample a group of $G$ candidate actions $\{a_j^i\}_{j=1}^G$ from the SFT policy $\pi_{\theta_{\text{SFT}}}(\cdot \mid \tau_{\leq(o^i)})$. The GRPO loss is formulated as:

$$
\mathcal{L}_{\text{GRPO}} = \mathbb{E}_{o^i \sim \mathcal{D}_{\text{OOD}}, \; \{a_j^i\}_{j=1}^G \sim \pi_{\theta_{\text{SFT}}}(\cdot \mid \tau_{\leq(o^i)})} \left\{ \frac{1}{G} \sum_{j=1}^G \frac{1}{|a_j^i|} \sum_{t=1}^{|a_j^i|} \right.
$$

$$
\min \left[ \frac{\pi_\theta(a_{j,t}^i \mid \tau_{\leq(o_t^i)})}{\pi_{\theta_{\text{SFT}}}(a_{j,t}^i \mid \tau_{\leq(o_t^i)})} \hat{A}_{j,t}, \text{clip} \left( \frac{\pi_\theta(a_{j,t}^i \mid \tau_{\leq(o_t^i)})}{\pi_{\theta_{\text{SFT}}}(a_{j,t}^i \mid \tau_{\leq(o_t^i)})}, \; 1-\epsilon, \; 1+\epsilon \right) \hat{A}_{j,t} \right]
$$

$$
\left. - \beta \, \mathbb{D}_{KL} \left[ \pi_\theta \, \| \, \pi_{\text{ref}} \right] \right\}, \tag{7}
$$

where $\hat{A}_{j,t}$ denotes the normalized advantage computed from the relative rewards within each sampled group, $|a_j^i|$ is the sequence length of the $j$-th sampled action, $\epsilon$ is the clipping threshold to ensure training stability, and $\beta$ controls the strength of KL divergence regularization. Through GRPO training, the DLM model achieves improved robustness against OOD actions while maintaining consistency with in-distribution behaviors established during SFT.

## 4 EXPERIMENT

In this section, we evaluate DLM against a range of offline multi-agent baselines. Since DLM adopts sequence modeling without explicit value functions, we focus comparisons on two categories: (1) value-based offline MARL algorithms that mitigate distributional shift through value pessimism or action regularization, and (2) imitation learning approaches trained via supervised objectives. Specifically, we compare against CFCQL (Shao et al., 2023), OMIGA (Wang et al., 2024), MACQL (Formanek et al., 2024) and TD3+BC (Kostrikov et al., 2021) as value-based baselines, and benchmark DLM against MADT (Meng et al., 2023) and BC (Syed et al., 2008) as sequence modeling and supervised learning baselines. Our evaluation spans all tasks in SMAC and a subset of tasks in SMACv2, covering easy, hard, and super hard settings that require decentralized coordination under partial observability. We report overall decision quality, analyze improvements in OOD robustness after preference optimization, and assess DLM's zero-shot generalization to unseen tasks. Full experimental setups, including hyperparameter selection and analysis, computational resources, and other implementation details, are provided in Appendix A.4, while additional experiments are presented in Appendix A.5.

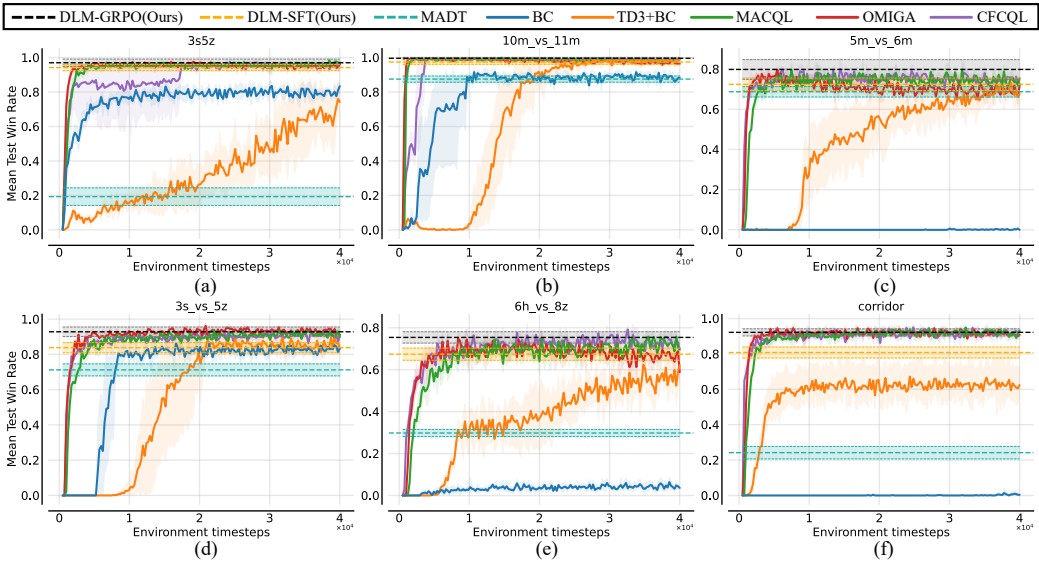

Figure 5: Performance comparison with baselines on representative SMAC tasks: (a)-(b) easy, (c)-(d) hard, and (e)-(f) super hard. Only the final test performance of DLM and MADT is reported.

### 4.1 OVERALL PERFORMANCE ON SMAC BENCHMARK

We evaluate DLM across 15 SMAC tasks. The evaluation focuses on mean test win rates as a measure of overall decision quality under decentralized partial observability. All baselines are trained on offline datasets collected following the procedure described in Sec. 3.1. The representative results are presented in Fig. 5, while the complete set of results can be found in Appendix Fig. 9. For fair evaluation, all experiments are conducted with five random seeds, and results are reported as means with a 95% confidence interval. Note that DLM-SFT and DLM-GRPO use a single model across all tasks, while other baselines are individually trained for each task.

Across all SMAC tasks, DLM-GRPO achieves the best average win rates with a single model, matching or surpassing strong value-based baselines such as OMIGA and CFCQL. DLM-SFT performs comparably to MACQL but remains slightly behind DLM-GRPO. TD3+BC and BC achieve reasonable results on easier tasks but generally fail to learn effective policies in harder tasks. MADT outperforms TD3+BC on several tasks but still struggles compared to DLM-SFT and value-based methods, although it consistently exceeds BC. Within imitation-based approaches, DLM significantly outperforms MADT and BC across most tasks. This can be attributed to DLM's dialogue-style sequence construction and multi-agent trajectory alignment, which

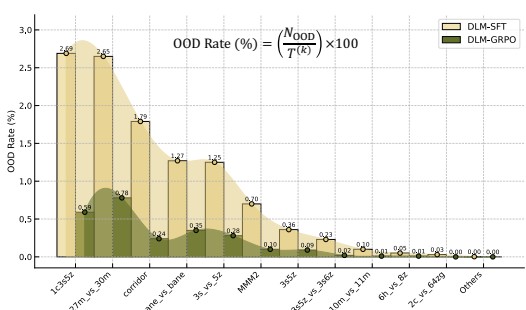

Figure 6: Comparison of OOD Rates on all SMAC tasks.

better preserve decision dependencies across agents and timesteps. Among value-based offline MARL methods, OMIGA and CFCQL achieve strong results. Despite not utilizing explicit value functions or environment rewards during training, DLM-GRPO achieves comparable or better results than OMIGA and CFCQL, highlighting the potential of lightweight preference optimization combined with autoregressive decision modeling. Finally, it is worth noting that DLM-SFT, trained on dialogue-style offline data without reward supervision, already matches MACQL, demonstrating the effectiveness of the DLM framework and its generalization ability. The consistent performance

improvement from DLM-SFT to DLM-GRPO further validates the role of simple preference optimization in enhancing robustness, mitigating OOD errors, and improving overall decision quality.

## 4.2 OOD Robustness Improvement via Preference Optimization

To evaluate the impact of preference optimization, we compare OOD action rates before and after applying GRPO, where OOD actions are defined as those violating environment constraints or deviating from offline behaviors (Sec.3.3). As shown in Fig. 6, DLM-SFT, despite its competitive decision quality, occasionally produces OOD actions due to limited data coverage and the absence of action masking. GRPO substantially reduces OOD rates across all tasks, enhancing robustness and execution stability. This reduction is especially pronounced in hard and super hard tasks, where a single OOD action can expose the agent to unseen states and trigger error cascades. A comparison between Fig. 4 and Fig. 6 reveals that high OOD rates correlate with sparse offline distributions (e.g., `1c3s5z`, `27m_vs_30m`, `corridor`), and these tasks also show degraded performance. In contrast, tasks with lower OOD occurrence typically yield higher win rates. These results confirm that preference optimization effectively mitigates OOD risks under limited data and improves generalization without additional reward modeling or environment interaction.

## 4.3 Zero-Shot Generalization to Unseen Tasks

To evaluate the generalization of DLM beyond its training distribution, we assess its zero-shot performance on unseen tasks from both SMAC and SMACv2. These tasks are excluded from training and introduce novel unit types, asymmetric team compositions, and coordination patterns. As reported in Tab. 1, the unified DLM-GRPO model achieves consistently strong win rates on representative tasks such as MMM, while maintaining competitive performance on more communication-intensive tasks. Although performance decreases in the most challenging SMACv2 tasks,

Table 1: Zero-shot performance comparison of MADT, DLM-SFT, and DLM-GRPO on unseen tasks. * communication tasks in SMAC (Wang et al., 2019); † SMACv2 tasks.

| Task | MADT | DLM-SFT | DLM-GRPO |
|---|---|---|---|
| 3s_vs_3z | $0.45 \pm 0.03$ | $0.71 \pm 0.02$ | $0.78 \pm 0.02$ |
| 3s_vs_4z | $0.73 \pm 0.02$ | $0.79 \pm 0.03$ | $0.82 \pm 0.02$ |
| 3m | $0.92 \pm 0.01$ | $0.90 \pm 0.04$ | $0.93 \pm 0.03$ |
| 8m | $0.69 \pm 0.02$ | $1.00 \pm 0.00$ | $1.00 \pm 0.00$ |
| 25m | $0.57 \pm 0.02$ | $0.98 \pm 0.01$ | $0.99 \pm 0.01$ |
| MMM | $0.85 \pm 0.02$ | $0.99 \pm 0.01$ | $1.00 \pm 0.00$ |
| 1o_10b_vs_1r* | $0.13 \pm 0.01$ | $0.57 \pm 0.01$ | $0.64 \pm 0.01$ |
| 1o_2r_vs_4r* | $0.11 \pm 0.04$ | $0.64 \pm 0.02$ | $0.69 \pm 0.02$ |
| protoss_5_vs_5† | $0.00 \pm 0.00$ | $0.59 \pm 0.05$ | $0.67 \pm 0.01$ |
| terran_5_vs_5† | $0.00 \pm 0.00$ | $0.64 \pm 0.01$ | $0.79 \pm 0.02$ |
| zerg_5_vs_5† | $0.00 \pm 0.00$ | $0.42 \pm 0.03$ | $0.58 \pm 0.02$ |

the variation aligns with task similarity to the training distribution: tasks closer to the training data yield higher generalization. Compared with MADT, both DLM-SFT and DLM-GRPO achieve better win rates across nearly all tasks, highlighting the advantage of dialogue-style sequence modeling. Overall, these results confirm that DLM can transfer decision behaviors to structurally novel environments, demonstrating scalability and robustness in zero-shot multi-agent settings.

## 5 Conclusion

In this paper, we present DLM, a scalable decision language model for offline multi-agent sequential decision-making cross tasks. By reformulating decision processes as dialogue-style sequence modeling, DLM bridges the gap between LLMs and decentralized decision problems. The two-stage training framework, consisting of SFT followed by GRPO, enables DLM to align with environment constraints, mitigate OOD errors, and generalize across tasks without relying on explicit rewards. Experiments on the SMAC benchmark show that DLM-SFT, trained solely on observations and actions, performs competitively with strong offline baselines. Building on this, DLM-GRPO further enhances robustness and decision quality, outperforming SOTA offline methods with a single unified model. Detailed analysis reveals that GRPO effectively reduces OOD action rates, particularly in complex tasks with limited data coverage. DLM also exhibits strong zero-shot generalization to unseen tasks, demonstrating scalability and adaptability. Overall, DLM provides a scalable solution to the long-standing generalization bottlenecks in offline MARL, laying the groundwork for building universal decision models that can be deployed in real-world embodied systems.

ETHICS STATEMENT

This work uses only simulated environments (SMAC and SMACv2) and does not involve human subjects, personal data, or sensitive content. All datasets are generated by agent–environment interaction in simulation, and no proprietary assets are redistributed. The research poses no direct societal or ethical risks beyond the standard concerns of deploying decision-making models in safety-critical domains, which is outside the scope of this paper. The authors declare no conflicts of interest.

REPRODUCIBILITY STATEMENT

We provide detailed descriptions of dataset construction (Sec. 3.1), model design (Sec. 3), and training objectives (Eqs. 2–7). Experimental settings, hyperparameters, and ablations are reported in Appendix A.4 and A.5. All results are averaged over five random seeds with 95% confidence intervals. Supplementary Material contains the corresponding code and configuration files.

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

# A APPENDIX

## A.1 DATASET CONSTRUCTION DETAILS

**Data Collection**  To ensure the credibility and reproducibility of our experimental results, we initially explored the use of publicly available offline MARL datasets. While several recent datasets, such as OG-MARL (Formanek et al., 2023), provide high-quality offline trajectories for MARL, they typically cover only a limited subset of SMAC (Samvelyan et al., 2019) tasks and do not support comprehensive multi-task evaluation. To address this limitation, we construct our own dataset following the data collection methodology of D4RL (Fu et al., 2020). Specifically, we adopt TGC-Net (Zhang et al., 2025b) as the behavior policy for data collection, due to its ability to achieve near-perfect performance across all SMAC tasks. For each task, we train TGCNet in an online setting and apply early stopping once the win rate exceeds 80%, using a test interval of 50000 steps. The resulting checkpoint is then used to interact with the environment and collect 4000 high-quality trajectories per task. Each collected trajectory includes standard components commonly used in offline MARL benchmarks: `actions`, `actions_onehot`, `avail_actions`, `filled`, `obs`, `reward`, `state` and `terminated`. As detailed in Sec.3.1, this process yields a diverse and consistent dataset spanning all 15 SMAC tasks. For DLM, the dataset is split into two subsets, one for SFT and the other for GRPO. Importantly, all baseline algorithms, including both value-based and imitation-based methods, are trained using the full dataset without any modifications. Therefore, although DLM employs a two-stage training procedure with a dataset split, it uses the same total amount of data as other baselines. All methods operate on an identical data distribution to ensure fairness and comparability.

**Dataset Quality**  To verify the reliability and quality of the collected trajectories, we conduct a quantitative analysis of the dataset. For each SMAC task, we divide the 4000 collected episodes evenly into two subsets of 2000 trajectories. This partition supports the two-stage training procedure of DLM, where one subset is used for SFT and the other for GRPO. We compute the average return and standard deviation for each subset by summing the per-step rewards within each episode and then aggregating statistics over 2000 trajectories. The results are presented in Tab. 2, which reports the mean $\pm$ standard deviation of episode returns for every task in both subsets, along with the overall average across all tasks. Although the maximum achievable return varies by task depending on episode length and reward sparsity, most of the collected trajectories yield returns consistent with the threshold used in our early-stopping strategy, where trajectory generation begins once TGCNet reaches a win rate of at least 80%. This confirms that the dataset meets the expected quality standard and is suitable for training reliable offline policies.

**Dialogue-Style Conversion**  For training DLM, we apply an additional transformation to the original offline dataset, converting it into a dialogue-style format suitable for multi-turn sequence modeling. In this process, we retain only the `obs` and `actions` fields at each timestep and organize them into observation–action pairs. Each pair is then verbalized as a natural language dialogue turn, as illustrated in Fig. 2 of the main text. We begin by constructing the *system prompt*, which corresponds to the system role in the chat interface and provides high-level scenario context. Since each SMAC task corresponds to a specific map, we generate map-specific instructions in the form: `"You are a strategic SMAC AI assistant on the _ map. Work with your team to complete the task."` This prompt guides the model to behave as a cooperative agent grounded in the given environment. Next, we generate the *observation prompt*, which corresponds to the user role in the chat interaction and encodes the agent's local observation at the current timestep. To construct this prompt, we assign each agent a unique ID and extract key features from its observation vector. The agent's own attributes, such as ID, unit type, health, and shield, are obtained from the `own_feats` field. Information about allied units within the agent's sight is retrieved from `ally_feats`, which includes their IDs, types, relative positions in X and Y coordinates, health, and shield. Likewise, the `enemy_feats` field provides corresponding information for visible enemies. These features are verbalized into a structured natural language description that captures the agent's local perspective. Finally, we construct the *action prompt*, which corresponds to the assistant role in the chat interaction and represents the agent's response based on its selected action. Since actions in the dataset are represented as discrete indices, we first decode each index based on the SMAC action mapping. For example, action 0 corresponds to `"no-op"`, action 1 to `"stop"`, and action 2 to `"move north one step"`. The decoded action is then

Table 2: Trajectory return (mean $\pm$ std) per SMAC task in the collected offline dataset. Each map contains 2000 trajectories divided equally into two subsets.

| Task | SFT Subset | GRPO Subset | Total Dataset |
|---|---|---|---|
| 2s_vs_1sc | $19.18 \pm 2.78$ | $19.17 \pm 2.79$ | $19.17 \pm 2.78$ |
| 2s3z | $19.89 \pm 0.89$ | $19.89 \pm 0.88$ | $19.89 \pm 0.89$ |
| 3s5z | $19.74 \pm 1.19$ | $19.76 \pm 1.08$ | $19.75 \pm 1.14$ |
| 1c3s5z | $19.98 \pm 0.44$ | $19.97 \pm 0.55$ | $19.97 \pm 0.50$ |
| 10m_vs_11m | $19.79 \pm 1.15$ | $19.72 \pm 1.35$ | $19.75 \pm 1.25$ |
| 2c_vs_64zg | $20.00 \pm 1.12$ | $19.99 \pm 1.14$ | $19.99 \pm 1.12$ |
| 5m_vs_6m | $18.28 \pm 3.74$ | $18.35 \pm 3.59$ | $18.34 \pm 3.66$ |
| bane_vs_bane | $20.00 \pm 0.00$ | $20.00 \pm 0.00$ | $20.00 \pm 0.00$ |
| 3s_vs_5z | $21.64 \pm 1.33$ | $21.62 \pm 1.39$ | $21.63 \pm 1.36$ |
| 8m_vs_9m | $19.07 \pm 2.57$ | $19.04 \pm 2.58$ | $19.06 \pm 2.58$ |
| 3s5z_vs_3s6z | $19.83 \pm 1.14$ | $19.84 \pm 1.07$ | $19.83 \pm 1.11$ |
| 27m_vs_30m | $19.17 \pm 2.20$ | $19.25 \pm 2.20$ | $19.21 \pm 2.20$ |
| 6h_vs_8z | $18.46 \pm 2.47$ | $18.45 \pm 2.50$ | $18.45 \pm 2.48$ |
| MMM2 | $19.41 \pm 1.96$ | $19.38 \pm 1.99$ | $19.40 \pm 1.98$ |
| corridor | $19.76 \pm 2.11$ | $19.69 \pm 2.22$ | $19.72 \pm 2.17$ |
| **Average** | $\mathbf{19.61 \pm 1.67}$ | $\mathbf{19.60 \pm 1.68}$ | $\mathbf{19.61 \pm 1.68}$ |

**ChatSMAC**

```
<|begin_of_text|>
<|start_header_id|>system<|end_header_id|>
You are a strategic SMAC AI assistant on the 2s_vs_1sc map.
Work with your team to complete the task. <|eot_id|>
...
<|start_header_id|>user<|end_header_id|>
Own feature, ID/T/H/S: 0/-/100.00%/62.50%.
Allies in sight, ID/T/X/Y/H/S: 1/-/0.65/0.00/100.00%/100.00%.
Enemies in sight, ID/T/X/Y/H/S:
  2/-/0.32/0.58/100.00%/0%. <|eot_id|>

<|start_header_id|>assistant<|end_header_id|>
attack ID 2. <|eot_id|>

<|start_header_id|>user<|end_header_id|>
Own feature, ID/T/H/S: 0/-/97.50%/62.50%.
Allies in sight, ID/T/X/Y/H/S: 1/-/0.65/0.00/100.00%/100.00%.
Enemies in sight, ID/T/X/Y/H/S:
  2/-/0.32/0.58/80.00%/0%. <|eot_id|>

<|start_header_id|>assistant<|end_header_id|>
attack ID 2. <|eot_id|>
...
```

Figure 7: An example trajectory from the ChatSMAC dataset formatted in chat format.

used as the assistant's response, completing the observation–action dialogue turn. Each complete turn is wrapped using the chat format adopted by LLaMA-3 (Grattafiori et al., 2024), ensuring compatibility with mainstream language models. This process produces the final dialogue-style dataset, which we refer to as ChatSMAC. It serves as the foundation for training DLM. Full examples of prompt construction and template specifications are provided in Fig. 7.

Figure 8: Screenshots of SMAC tasks at different difficulty levels: (a) `2s3z` (easy), (b) `2c_vs_64zg` (hard), and (c) `corridor` (super hard).

Table 3: SMAC task difficulty classification.

| Difficulty Level | Task |
|---|---|
| Easy | 2s_vs_1sc |
|  | 2s3z |
|  | 3s5z |
|  | 1c3s5z |
|  | 10m_vs_11m |
| Hard | 2c_vs_64zg |
|  | 5m_vs_6m |
|  | bane_vs_bane |
|  | 3s_vs_5z |
|  | 8m_vs_9m |
| Super Hard | 3s5z_vs_3s6z |
|  | 27m_vs_30m |
|  | 6h_vs_8z |
|  | MMM2 |
|  | corridor |

## A.2 DETAILS ABOUT BENCHMARKS

**SMAC Overview** The SMAC (Samvelyan et al., 2019) benchmark, built upon the StarCraft II engine, has been widely used for evaluating cooperative multi-agent reinforcement learning. It focuses on micromanagement tasks where each agent controls a single unit and must coordinate with others under partial observability and sparse global rewards. Recently, SMACv2 (Ellis et al., 2023) was introduced to address the limitations of SMAC by introducing additional randomness, restricting agents' field of view, and providing more diverse and challenging tasks, making it a more rigorous testbed for assessing generalization, robustness, and scalability of multi-agent learning methods.

**Task Grouping** SMAC tasks are commonly divided into three difficulty levels: easy, hard, and super hard. This classification is based on factors such as unit types, asymmetry between teams, and the complexity of required coordination. Easy tasks typically involve symmetric unit compositions and can often be solved with basic strategies. In contrast, hard and super hard tasks introduce heterogeneous units, asymmetric team settings, and demand more sophisticated tactics such as precise positioning, focus firing, and kiting. Representative examples of each difficulty level are illustrated in Fig. 8, highlighting the increasing complexity across categories. A full list of task groupings by difficulty is summarized in Tab. 3.

**Observation and Action Spaces** In each SMAC task, agents receive low-dimensional local observations that encode information about nearby allies, enemies, and the agent's own state. The observation space has a fixed dimensionality, but the content depends on the number of visible units, reflecting the partial observability of the environment. The action space is discrete and includes primitive operations such as moving in four directions, attacking or healing specific enemies

or allies, stopping, and executing a no-operation. Notably, the set of available actions is dynamic, determined by the agent's current local visibility and unit-specific constraints.

**Choice of Multi-Task Benchmarks** We adopt SMAC (Samvelyan et al., 2019) and its extension SMACv2 (Ellis et al., 2023) as the primary benchmarks for evaluating DLM in a multi-task setting. Following prior work such as MADT (Meng et al., 2023), each task within SMAC is conventionally treated as a distinct task, since maps differ in agent types and numbers, available abilities, team compositions, and coordination requirements. This diversity spans simple symmetric battles to highly asymmetric matchups that demand fine-grained cooperation, thereby aligning with common definitions of multi-task reinforcement learning as learning across a distribution of environments with varied state/action spaces and task goals. Compared with SMAC, SMACv2 introduces additional randomness, restricted fields of view, and more heterogeneous unit compositions, which further increase task variability and difficulty. Together, SMAC and SMACv2 provide a scalable and reproducible platform where heterogeneous cooperative tasks can be systematically evaluated under a unified framework, making them suitable for assessing the generalization capacity of large decision models like DLM.

### A.3 Motivation Behind the Design

A central component of DLM is the representation of multi-agent trajectories as sequences of observation–action pairs:

$$\tau^{(k)} = \left( \left\{ (o_t^{(k),i}, \, a_t^{(k),i}) \mid t = 1, \ldots, T^{(k)} \right\} \right)_{i=1}^{N}, \tag{8}$$

where $o_t^{(k),i}$ and $a_t^{(k),i}$ denote the local observation and executed action of agent $i$ at timestep $t$ in the $k$-th trajectory, and $T^{(k)}$ is the episode length. This design is motivated by a careful analysis of the limitations of BC in multi-agent settings.

**Limitation 1: Absence of Inter-Agent Information** Conventional BC learns a local policy $\pi^i(o^i) = p(a \mid o^i)$ by mapping the agent's private observation $o^i$ to an action $a$, while ignoring critical dependencies on other agents' information. In cooperative multi-agent tasks, a common idealization is to treat all agents as components of a single joint agent operating over the full global state $s$, in which case the optimal policy is defined as $\pi^i(s) = p(a \mid s)$. However, under partial observability, the agent's local observation $o^i$ may correspond to multiple possible global states, causing $p(a \mid o^i)$ to become a weighted mixture over the optimal policies for different $s$. This discrepancy leads to a mismatch between $p(a \mid o^i)$ and the true optimal policy $p(a \mid s)$. Formally, we can express this mismatch as:

$$p(a \mid o^i) = \sum_s p(a \mid s) \cdot p(s \mid o^i), \tag{9}$$

where the posterior $p(s \mid o^i)$ represents a distribution over global states consistent with $o^i$. When $p(s \mid o^i)$ has high entropy, the resulting policy becomes a blurred mixture, leading to suboptimal actions. Now consider conditioning on the joint observations of all agents $(o^1, \ldots, o^n)$. The corresponding policy is:

$$p(a \mid o^1, \ldots, o^n) = \sum_s p(a \mid s) \cdot p(s \mid o^1, \ldots, o^n). \tag{10}$$

From the information-theoretic property that conditioning reduces entropy (Cover & Thomas, 2006), we have:

$$\mathcal{H}(s \mid o^i) \geq \mathcal{H}(s \mid o^1, \ldots, o^n), \tag{11}$$

which implies that the posterior $p(s \mid o^1, \ldots, o^n)$ is more concentrated than $p(s \mid o^i)$.

As a result, the weighted average in Eq. 10 more closely approximates the true optimal policy $p(a \mid s)$ than Eq. 9. This analysis highlights that incorporating full agent trajectories, as in our dialogue-style formulation, reduces the representation gap and leads to higher-quality decision making compared to conventional BC.

**Limitation 2: Lack of Temporal Dependency**  BC also suffers from ignoring temporal dependencies by treating each timestep independently, i.e., modeling the policy as $p(a_t^i \mid o_t^i)$ without incorporating past observations or actions. However, in partially observable environments, the current observation $o_t^i$ alone is generally insufficient to infer the true latent state of the environment. As a result, this memoryless policy lacks the contextual information required for strategic reasoning over time.

To formalize this limitation, consider that the optimal policy in a Dec-POMDP depends on the full action-observation history $h_t^i = (o_1^i, a_1^i, \dots, o_{t-1}^i, a_{t-1}^i, o_t^i)$. The true optimal policy is therefore:

$$\pi_{\text{opt}}^i = p(a_t^i \mid h_t^i), \tag{12}$$

whereas BC approximates this as:

$$\pi_{\text{BC}}^i = p(a_t^i \mid o_t^i). \tag{13}$$

Applying the data processing inequality (Cover & Thomas, 2006), we know that:

$$I(a_t^i; h_t^i) \geq I(a_t^i; o_t^i), \tag{14}$$

where $I(\cdot\,;\cdot)$ denotes mutual information. This inequality highlights that conditioning on full history provides strictly more information about the optimal action than conditioning on $o_t^i$ alone.

Thus, ignoring historical context reduces the model's capacity to learn strategies that rely on long-term planning, multi-agent coordination, or temporal disambiguation. Such strategies are frequently required in complex tasks, including kiting, flanking maneuvers, or delayed action execution. DLM addresses this limitation by formulating decision-making as an autoregressive sequence modeling problem. By retaining the full sequence of past observation–action pairs as dialogue history, the model can effectively leverage long-range temporal dependencies to make more informed and coherent decisions.

**Design Motivation of DLM**  To address these limitations, DLM reformulates the decision process as a dialogue-style sequence modeling problem. Instead of fitting per-timestep policies independently, it models the entire trajectory as a structured autoregressive sequence of $(o, a)$ pairs. This design allows:

- **Contextual Encoding**: Inter-agent relationships are captured implicitly in the structured dialogue, where each agent's input includes both its own and nearby agents' attributes, encoded in natural language.
- **Temporal Dependency**: By modeling decisions autoregressively, the model naturally learns from the accumulated context of previous observations and actions, thus capturing history without explicit recurrence.
- **CTDE Compatibility**: Representing trajectories in language format naturally supports the CTDE paradigm. Each dialogue-style trajectory encodes the full decision process of an individual agent, while maintaining access to global information across agents during training. This enables the model to learn coordinated strategies centrally, yet make decisions based solely on local observations during execution.

In summary, the design of Eq. 8 serves as a unified interface that preserves agent-level autonomy while enabling temporally and contextually grounded decision-making. This structure is essential for bridging the gap between language models and multi-agent sequential decision processes.

### A.4 TRAINING AND IMPLEMENTATION DETAILS

**Supplementary Results for Sec. 4.1**  In Sec. 4.1, we reported results on six representative tasks from the SMAC benchmark. Here we provide results for the remaining nine tasks in Fig. 9. Taken together with Fig. 5, the overall trends across all tasks remain consistent with the earlier analysis.

**Baseline Implementation**  We compare DLM with a set of representative offline MARL baselines, which fall into two major categories: value-based methods and imitation-based methods. Among value-based methods, TD3+BC (Kostrikov et al., 2021) applies a conservative value estimation strategy by combining actor-critic learning with behavior cloning regularization, aiming to

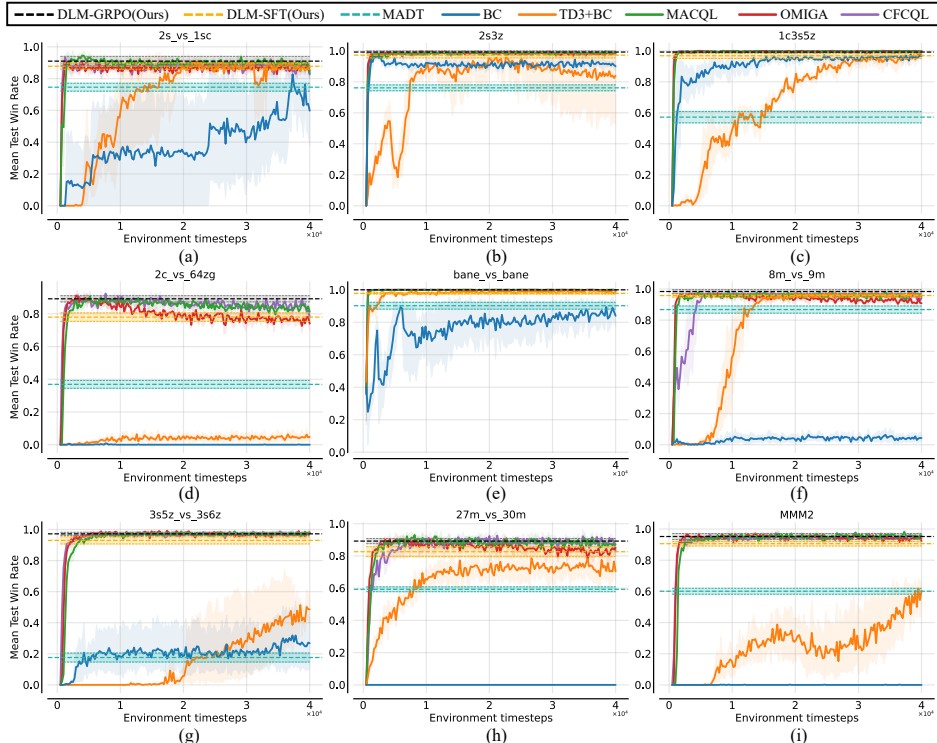

Figure 9: Performance comparison with baselines on remaining SMAC tasks: (a)-(c) easy, (d)-(f) hard, and (g)-(i) super hard. Only the final test performance of DLM and MADT is reported.

reduce overestimation and improve stability. MACQL (Formanek et al., 2024) extends the idea of CQL (Kumar et al., 2020) to the multi-agent setting by incorporating joint action masking under the CTDE paradigm. OMIGA (Wang et al., 2024) improves upon previous methods by incorporating global-to-local value shaping to better guide decentralized agents during training. CFCQL (Shao et al., 2023) further enhances robustness by introducing counterfactual regularization at the agent level, allowing better credit assignment under partial observability. On the imitation-based side, BC (Syed et al., 2008) directly learns policies via supervised learning from offline action labels without any value estimation. MADT (Meng et al., 2023) formulates decision-making as an autoregressive sequence prediction problem and leverages return-conditioning to generalize across tasks and agent configurations, although it requires environment interaction for online fine-tuning.

For implementation, we build all value-based baselines on top of the EPyMARL framework (Papoudakis et al., 2020), which is designed for flexible MARL experimentation. For imitation-based baselines, we adapt publicly available implementations released by the original authors. Where no official code is available, we reproduce the algorithms based on their published descriptions and validate the implementations by replicating reported performance. All baseline methods are trained on our collected offline dataset to ensure consistency and fairness in comparison with DLM.

**Implementation of DLM** DLM is implemented using the Hugging Face Transformers library and integrated with standard multi-agent datasets. We adopt LLaMA-3.2-1B as the pre-trained language model backbone. The training process is divided into two stages: SFT and preference-based alignment via GRPO. In the SFT stage, we perform full-parameter fine-tuning on the first half (2000 trajectories) of our collected offline dataset. All 15 SMAC tasks are mixed and fed into the model in a single training run. We choose not to use parameter-efficient methods like LoRA (Hu et al., 2021) at this stage because our experiments show that, although LoRA can reduce training cost, it limits the model's representation capacity when dealing with diverse multi-task data. Given the small model size (1B parameters), full fine-tuning ensures sufficient capacity to fully adapt to all environments. Moreover, training on all tasks jointly avoids catastrophic forgetting that may occur if the model is fine-tuned sequentially on different tasks. In the GRPO stage, we freeze the base model from SFT

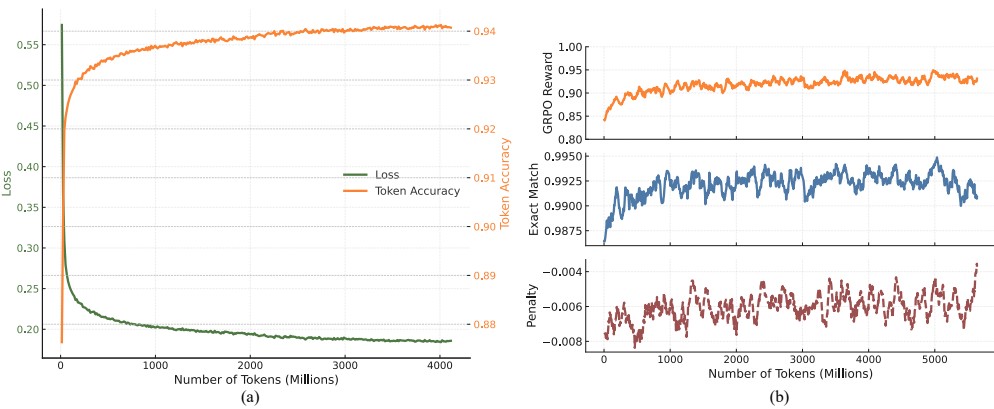

Figure 10: Training curves of DLM. (a) DLM-SFT: cross-entropy loss and token accuracy over tokens. (b) DLM-GRPO: reward (top), exact match rate (middle), and penalty (bottom).

and apply LoRA-based fine-tuning on the remaining half of the dataset (another 2000 trajectories). This stage focuses on reducing OOD errors by leveraging lightweight executability-based reward signals. Using LoRA here allows efficient alignment while preserving the general decision-making capability acquired during SFT. This design balances robustness and generalization and enables DLM to adapt without overwriting previously learned behaviors. Fig. 10 summarizes the learning dynamics. In subplot (a), we observe a consistent decline in SFT loss alongside a steady increase in token-level accuracy, which eventually reaches 94%, indicating that DLM-SFT can effectively learn to reproduce behaviors from the offline trajectories across all tasks. In subplot (b), during the GRPO stage, the preference reward steadily improves. At the same time, the exact match rises suggesting that the model's outputs increasingly match preferred actions even in challenging or OOD-prone observations. The penalty term also gradually approaches zero, indicating a decreasing frequency of invalid or infeasible actions. These trends validate the effectiveness of our two-stage design in first imitating multi-agent behavior and then refining it through preference optimization.

**Hyperparameter Settings**    All value-based baselines, including CFCQL, OMIGA, MACQL, and TD3+BC, are implemented within the EPyMARL framework. We follow the default hyperparameter settings provided in the official implementations or corresponding papers to ensure reproducibility and fairness. For example, buffer sizes are set to 5000, the learning rate is fixed at 5e-4, and $\epsilon$-greedy exploration is applied with $\epsilon$ linearly annealed from 1.0 to 0.05 over 50,000 steps. All imitation-based baselines, including MADT and BC, are trained using the same offline dataset with their original configurations where available, ensuring consistent training conditions across all methods. Key training hyperparameters for DLM are summarized in Tab. 4. All experiments are conducted on 8 NVIDIA L40 GPUs.

**Hyperparameter Tuning**    We adopt systematic strategies to tune the hyperparameters of DLM and ensure fair comparison with baseline algorithms. For all value-based methods (CFCQL, OMIGA, MACQL, and TD3+BC), we use the official hyperparameter settings from their original papers or public implementations. These configurations have been validated across the SMAC benchmark. For DLM, we perform controlled hyperparameter tuning on both the SFT and GRPO stages. Specifically, we tune the learning rate, context length, LoRA rank, KL divergence coefficient ($\beta$), PPO clipping threshold ($\epsilon$), and sampling parameters (top-$k$, top-$p$). The search process is guided by performance on a held-out subset of SMAC tasks (e.g., 3s5z, 10m_vs_11m, and MMM2). Considering the substantial computational cost of training across all tasks, we limit the search to a representative subset to efficiently explore the hyperparameter space. For each hyperparameter combination, we train the model under three different random seeds and select the configuration that achieves the highest average win rate across these runs.

Unlike prior methods that require task-specific tuning, DLM uses a unified set of hyperparameters across all 15 SMAC tasks. This design choice enhances generalization and prevents overfitting to

Table 4: Hyperparameters used for DLM training.

| Hyperparameter | Value |
|---|---|
| *DLM-SFT* | |
| Learning rate | 2e-5 |
| Batch size | 8 |
| Gradient accumulation steps | 8 |
| Max length | 1024 tokens |
| Epochs | 2 |
| Packing | True |
| *DLM-GRPO* | |
| LoRA rank ($r$) | 128 |
| LoRA $\alpha$ | 256 |
| Batch size | 8 |
| Number of generations per sample | 4 |
| Epochs | 2 |
| Learning rate | 5e-5 |
| KL coefficient ($\beta$) | 0.1 |
| PPO clipping threshold ($\epsilon$) | 0.2 |
| Top-$k$ / Top-$p$ sampling | 50 / 0.95 |

any particular map. The final selected hyperparameters are summarized in Tab. 4, and all reported results are obtained using this fixed configuration without further tuning.

**Computational Cost** We analyze the computational cost of all algorithms in terms of mean training duration per difficulty level and total GPU hours across all SMAC tasks. As shown in Fig. 11(a), we observe two key trends. First, more complex algorithms, such as OMIGA and CFCQL, generally require longer training time, particularly on easy and hard tasks. Second, task difficulty tends to correlate positively with training duration, as more challenging environments typically demand longer convergence. An exception is observed with BC, which exhibits unusually high training time even on some easy tasks. A plausible explanation is that BC struggles to converge in these cases, and as a result, it often interacts with the environment until reaching the maximum episode length during each training iteration, thereby increasing the overall runtime.

In contrast, as shown in Fig. 11(b), DLM exhibits the lowest overall computational cost among all evaluated methods, despite its two-stage training pipeline. Specifically, DLM-SFT and DLM-GRPO require approximately 60 and 70 GPU-hours respectively when trained across the full benchmark. Notably, this is significantly lower than the cumulative training cost of value-based baselines, which must be trained independently for each task. As such, the total cost reported for these baselines is obtained by summing their per-task training durations. In comparison, DLM benefits from its unified text-based formulation, enabling multi-task generalization via a single language model. This eliminates the need for repeated training or task-specific value function updates, resulting in substantial computational savings. These properties highlight DLM's efficiency and scalability in large-scale offline MARL settings, offering a compelling balance of performance and resource efficiency.

## A.5 ADDITIONAL EXPERIMENTAL RESULTS

### A.5.1 ABLATION ON GRPO USAGE

To assess the necessity of second-stage preference optimization (GRPO), we conduct an ablation study where all 4000 trajectories are used solely for SFT, omitting GRPO updates. We compare three configurations: DLM-SFT trained on 2000 trajectories, full-data SFT trained on all 4000 trajectories, and the full DLM pipeline combining SFT and GRPO. Tab. 5 reports win rates and OOD action rates across representative SMAC tasks.

We observe that increasing the training data from 2000 to 4000 trajectories can lead to marginal improvements in some challenging environments such as 27m_vs_30m and MMM2, where data cov-

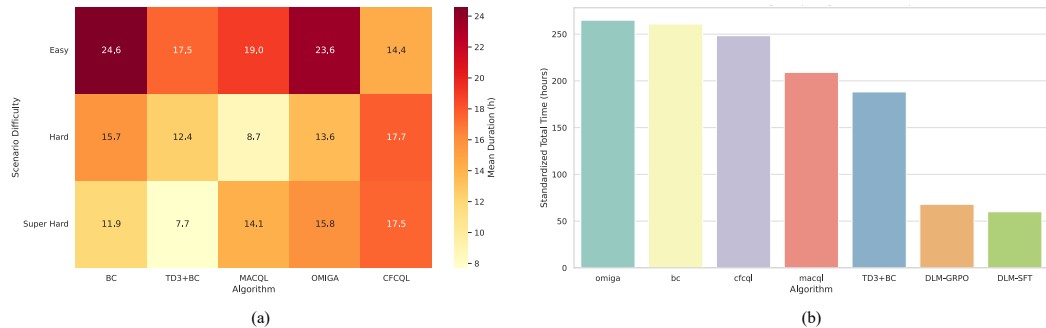

(a)                                                                  (b)

Figure 11: Computational cost comparison. (a) Average training time per difficulty level for each baseline. (b) Total standardized GPU hours across all tasks.

Table 5: Comparison of win rate (%) and OOD rate (%) across three configurations: DLM-SFT (2000), full-data DLM-SFT (4000), and DLM (SFT+GRPO).

| Task | Win Rate (%) | | | OOD Rate (%) | | |
|---|---|---|---|---|---|---|
| | SFT (2000) | SFT (4000) | DLM | SFT (2000) | SFT (4000) | DLM |
| 2s3z | 98.5 | 98.4↓ | **99.7**↑ | 0.00 | 0.00 | 0.00 |
| 3s5z_vs_3s6z | 84.2 | 81.2↓ | **89.5**↑ | 0.23 | 0.19↓ | **0.02**↓ |
| 27m_vs_30m | 80.1 | 81.9↑ | **84.6**↑ | 2.65 | 2.41↓ | **0.78**↓ |
| MMM2 | 90.8 | 91.7↑ | **97.3**↑ | 0.70 | 0.58↓ | **0.10**↓ |
| 1c3s5z | 97.7 | **99.0**↑ | 98.8↑ | 2.69 | 2.23↓ | **0.59**↓ |

erage helps mitigate underfitting. For example, in MMM2, win rate increases from 90.8% to 91.7%. However, these gains are modest and come with persistently high OOD rates, indicating that naive data scaling does not eliminate OOD behavior. In easier tasks such as 2s3z, full-data SFT provides only limited benefit and may even induce slight overfitting, as shown by the stagnating or slightly decreased win rates. In contrast, the full DLM configuration consistently improves both win rate and OOD robustness. These results confirm that data quantity alone is insufficient to address OOD generalization in multi-agent settings. This supports the analysis in Sec. 3.1, that increased data alone cannot resolve OOD issues, and explicit alignment mechanisms like GRPO are essential.

### A.5.2 ABLATION ON OOD FILTERING

We further examine the impact of executability-based filtering before applying GRPO. In the ablation setting, GRPO is directly applied to all remaining trajectories without removing OOD-prone samples identified during the SFT stage. Empirically, we observe significant training instability: unlike the clean and steadily improving curves shown in Fig. 10(b), reward signals, exact match rates, and penalties fluctuate erratically throughout training. This behavior suggests that the model struggles to converge when exposed to a mixture of already-correct and severely misaligned samples. Furthermore, in tasks where DLM-SFT already achieves high performance (e.g., 2s3z), the absence of filtering introduces noisy gradient updates that degrade accuracy and increase training time. These results underscore the necessity of targeted optimization and validate the role of OOD filtering in enabling stable and efficient GRPO alignment.

### A.5.3 SCALABILITY TO ADDITIONAL BENCHMARKS

To further evaluate the scalability of DLM beyond SMAC, we expand training and testing to additional multi-agent benchmarks. Following the dataset construction methodology in Sec. 3.1, we collect dialogue-style offline datasets specifically from two tasks: Hallway:4×6×10 (Wang et al., 2019) and LBF:11×11-6p-4f (Papoudakis et al., 2020). Based on these two datasets, we train a DLM following the methodology described in this paper. Tab. 6 summarizes the results, where DLM-GRPO consistently outperforms or matches strong baselines across both benchmarks.

Table 6: Performance on additional training benchmarks.

| Task | DLM-GRPO | DLM-SFT | MADT | BC | TD3+BC | MACQL | OMIGA | CFCQL |
|------|----------|---------|------|-----|--------|-------|-------|-------|
| Hallway:4×6×10 | $0.83 \pm 0.04$ | $0.79 \pm 0.05$ | $0.77 \pm 0.02$ | $0.18 \pm 0.05$ | $0.16 \pm 0.03$ | $0.58 \pm 0.17$ | $0.70 \pm 0.06$ | $0.64 \pm 0.16$ |
| LBF:11×11-6p-4f | $0.96 \pm 0.02$ | $0.91 \pm 0.03$ | $0.85 \pm 0.09$ | $0.28 \pm 0.06$ | $0.30 \pm 0.05$ | $0.69 \pm 0.08$ | $0.85 \pm 0.07$ | $0.77 \pm 0.09$ |

Beyond training benchmarks, we also assess zero-shot transfer to additional tasks. On Hallway:6×6, DLM-GRPO achieves a win rate of $0.76 \pm 0.07$, while on LBF:20×20-10p-6f it achieves $0.69 \pm 0.05$. These results indicate that DLM maintains strong generalization across structurally diverse environments without task-specific adaptation.

## A.6 LIMITATIONS AND POSSIBLE NEGATIVE SOCIETAL IMPACTS

One limitation of our approach is the need to convert structured trajectory data into natural language dialogue format, which introduces additional preprocessing steps. Although this transformation requires some effort and time, it is a one-time process and remains manageable in practice. We find that the benefits of enabling compatibility with language models outweigh the modest cost of this data preparation phase.

## A.7 THE USE OF LARGE LANGUAGE MODELS

This paper has used large language models (LLMs) solely for language polishing, grammar checking, and improving clarity of presentation. No LLMs were involved in designing the research ideas, developing algorithms, conducting experiments, or analyzing results. All technical contributions, datasets, experiments, and conclusions were produced entirely by the authors.

