# OpenReview forum: "DLM: A Scalable Decision Language Model for Multi-Agent Sequential Decision in SMAC Tasks"
_ICLR.cc/2026/Conference — Submitted to ICLR 2026_

### Official Review · Reviewer_i8oP · 2025-10-25

**Soundness:** 2
**Presentation:** 2
**Contribution:** 2
**Rating:** 4
**Confidence:** 4

**Summary:**

The paper reformulates multi-agent sequential decision-making as a dialogue-style sequence modeling problem. Specifically, it converts observations and actions into textual “chat” sequences and trains a pre-trained LLM in two stages: DLM-SFT — supervised fine-tuning on text-converted trajectories; DLM-GRPO — reinforcement alignment using a lightweight executability reward on out-of-distribution (OOD) or inconsistent samples to reduce invalid actions and improve cross-task generalization. Experiments are conducted on SMAC and SMACv2 benchmarks, demonstrating improved performance over baseline policy and imitation learning methods. The paper also provides implementation details for the data generation pipeline, including the use of a high-performing policy (TGCNet) to collect trajectories after reaching a certain win rate threshold, and reports metrics such as win rate, OOD rate, and wall-clock efficiency.

**Strengths:**

1. The paper clearly specifies the SFT and GRPO loss functions, hyperparameters, sampling strategies (top-k, top-p), and LoRA setup. It also provides training details such as the number of seeds, confidence intervals, and evaluation metrics, which aid reproducibility.

2. Experiments cover multiple levels of SMAC and SMACv2 tasks, systematically reporting both in-distribution and out-of-distribution settings, as well as comparisons with imitation and RL-based baselines under a unified evaluation metric.

3. The data collection and text transformation pipeline is described in sufficient detail, including the use of behavioral policy thresholds, number of trajectories per task, tokenization templates, and dialogue formatting for multi-agent observations and actions.

**Weaknesses:**

1. OOD samples are defined as cases where the model’s action differs from the data action or is not executable. However, the GRPO reward simultaneously encourages exact matching to the data action. This setup risks penalizing alternative yet valid actions, effectively aligning the model to the behavior policy rather than improving true OOD robustness.

2. The reward depends on the environment-specific availability mask (A_avail) and equality to the data action. This tight coupling makes it unclear how the same mechanism would apply to continuous control tasks or domains without a discrete action mask.

3. All training trajectories are generated by a single strong behavior policy (TGCNet) after reaching high win rates. This creates a strong bias toward that policy’s distribution and limits the diversity of the dataset. As a result, the model may merely imitate a specific strategy rather than learning to generalize to unseen dynamics or alternative cooperative behaviors.

4. DLM is trained as a unified model covering multiple tasks simultaneously, whereas most baselines are trained per task. This setup reduces DLM’s total training cost per task but may also obscure fairness — the unified model benefits from shared representations while the baselines lack such cross-task regularization. The paper does not clarify whether hyperparameter tuning, action masking, or data splits were matched across methods.

5. The conversion of numerical observations and discrete actions into natural language templates is central to the method, yet the paper does not explore how different verbalization schemes, numeric precision, or prompt lengths affect performance or stability. The absence of such ablations leaves uncertainty about robustness to prompt engineering choices.

**Questions:**

1.  Beyond exact equality to the data action, have the authors considered equivalence classes or value-based thresholds to recognize different but equally valid actions and avoid penalizing them as OOD?

2.  How would the proposed reward formulation adapt to environments without discrete action masks or where the available-action set is undefined (e.g., continuous control)?

3.  Can the authors report performance under variations in text templates (e.g., shorter vs. longer descriptions, rounding precision of numeric states, alternative tokenizations)?

4.  How does model performance and memory footprint change with longer horizons and a larger number of agents when dynamic token packing or truncation is modified?

5.  What is the quantitative impact of invalid-action resampling and masking on success rate and OOD ratio? Have the authors ablated this mechanism to isolate its contribution?

6.   Among all OOD cases, what proportion arises from *illegal* actions versus *legally executable but inconsistent* actions? Does GRPO primarily mitigate one class or both?

7.  Under the same text-based representation, how would a return-conditioned transformer (RTG-style) or advantage-conditioned model perform compared to GRPO alignment?

8.  Could the authors provide results where all baselines are also trained as a single multi-task model, or conversely, where DLM is trained per task, to better assess fairness?

9. Have the authors tested DLM’s zero-shot or few-shot performance on domains that differ in both observation and action semantics (e.g., cooperative card or robot tasks)?

10. How does DLM behave when the textual observation includes noise, missing values, or shuffled field order? Are there any mechanisms to maintain safe and valid actions under such perturbations?

**If you address my concerns, I will consider raising my score.**

---

### Official Review · Reviewer_cTP3 · 2025-10-30

**Soundness:** 3
**Presentation:** 2
**Contribution:** 3
**Rating:** 4
**Confidence:** 5

**Summary:**

1.	This paper presents the Decision Language Model (DLM), a novel framework that leverages pre-trained Large Language Models (LLMs) for offline multi-agent reinforcement learning (MARL).
2.	The core innovation is the reformulation of multi-agent decision-making as a dialogue-style sequence modeling problem. The proposed two-stage training pipeline—Supervised Fine-Tuning (SFT) followed by Group Relative Policy Optimization (GRPO)—effectively addresses key challenges in the field, including representation mismatch, Centralized Training with Decentralized Execution (CTDE) incompatibility, and individual reward scarcity.
3.	The authors provide extensive experimental results on the challenging SMAC and SMACv2 benchmarks, demonstrating that DLM achieves state-of-the-art performance with a single, unified model and exhibits strong zero-shot generalization.

**Strengths:**

1.  The idea of converting multi-agent trajectories into a natural language dialogue is elegant and powerful. It effectively tackles the representation mismatch problem by leveraging the LLM's inherent ability to handle diverse inputs through tokenization. The dialogue format naturally encapsulates the CTDE paradigm, preserving inter-agent context during training while allowing for decentralized execution.
2. The paper does an excellent job of identifying and systematically addressing three critical bottlenecks in multi-task offline MARL. Representation Mismatch is  solved via verbalization of observations and actions; CTDE Incompatibility is Solved via the dialogue-style trajectory representation.  Reward Scarcity & Credit Assignment is Solved by fine-tuning a pre-trained LLM (which can perform implicit credit assignment via temporal attention) and entirely discarding reward signals during SFT. This is a bold and effective design choice.
3.  The experimental section is thorough and convincing. DLM's performance is rigorously evaluated against a wide range of strong baselines (both value-based and imitation-based) across the full spectrum of SMAC difficulty levels.
4.  A notable advantage of DLM is its lower total computational cost compared to task-specific value-based methods, despite its two-stage training. This scalability is a significant practical benefit.

**Weaknesses:**

1.  While the paper argues that the pre-trained LLM provides a strong prior for decision-making, the experiments do not fully isolate its contribution. An ablation study comparing DLM initialized with a pre-trained LLM versus a randomly initialized Transformer of the same size would help quantify the value of the linguistic prior versus the architectural benefits of the Transformer itself.
2.  The current work uses the relatively small LLaMA-3.2-1B model. It is unclear how the approach would scale to larger LLMs in terms of performance and computational requirements. Furthermore, while SMAC is a standard benchmark, the actions and observations are still within a constrained, grid-like world. Testing on environments with richer, more natural language-based dynamics (e.g., text-based games or more complex simulators) would further validate the "language model for decision" premise.
3. The handcrafted executability reward for GRPO, while simple and effective, is quite sparse. It primarily penalizes invalid actions but does not actively guide the policy toward optimal behavior. The performance gains likely stem from avoiding catastrophic failures. Exploring denser, task-oriented heuristic rewards (even if still lightweight) could potentially lead to further performance improvements.
4.  For reproducibility, it is crucial that the constructed ChatSMAC dataset—the dialogue-formatted version of the offline data—is made publicly available. The paper should explicitly state its availability.

**Questions:**

1.  How does the model's performance and training stability change if the dialogue history is truncated more aggressively? Is there a point where reducing context length significantly harms performance, particularly in long-horizon tasks?
2.  The GRPO stage uses a KL penalty w.r.t. the SFT model as a reference. Did you experiment with using the original pre-trained LLM as the reference model instead, and if so, what were the results?
3.  Have you observed any failure modes or specific task types where the dialogue formulation or the LLM-based approach consistently underperforms compared to traditional value-based MARL methods?

---

### Official Review · Reviewer_sC1f · 2025-10-30

**Soundness:** 1
**Presentation:** 2
**Contribution:** 1
**Rating:** 2
**Confidence:** 4

**Summary:**

This manuscript proposes a new framework called the "Decision Language Model" (DLM), aiming to refactor the Offline MARL problem into a conversational sequence prediction problem. The authors aim to address the limitations of traditional MARL methods in generalization, particularly due to the fixed observation format and action space. The method employs a two-stage training pipeline including SFT and GRPO. The authors claim that the DLM framework addresses several key challenges in offline MARL, including representation mismatch, incompatibility between CTDE, and individual reward scarcity.

**Strengths:**

1. While serializing decision problems is not original, specifically formulating MARL as a “dialogue” is methodologically concise. The “verbalizing” of SMAC’s state vectors is a clever engineering design that injects inductive bias and effectively leverages the ability of pre-trained LLMs to process structured text. Its design for supporting CTDE (encoding information about allies and enemies as context into the cues) is also well-considered.
2. The authors constructed and documented a new large-scale dataset, “ChatSMAC,” in detail. Appendix A, with its detailed information on data collection (using TGCNet with a win rate >80%) and transformation process, is valuable to the community and will contribute to future research on LLM-MARL.

**Weaknesses:**

1. In Appendix A.1, the authors disclose their data collection methodology. They used a state-of-the-art online MARL agent (TGCNet) and only began collecting trajectories after it achieved a win rate >80%. By definition, this is an expert or near-expert dataset. However, The core challenge of the entire field of "offline RL" (e.g., as defined by the D4RL benchmark) is how to learn policies from suboptimal, mixed-quality, and unstructured datasets. A key task of offline RL is to "stitch" useful behavioral fragments from trajectories of varying quality using reward signals. It means the SFT stage of DLM is not "offline RL". It is an advanced, large-scale behavior cloning (BC) or imitation learning (IL) method, rendering the benchmark comparisons in the paper invalid. On a purely expert dataset, IL will almost always outperform Offline RL because IL's task (mimicry) is far simpler than Offline RL's task (pessimistic optimization + mimicry).
2. The paper claims its “conversational” sequence representation and “lightweight” OOD processing are innovative, but completely ignores the work most relevant to these two aspects: Gato (Reed et al. 2022)  and SayCan (Ahn et al. 2022).

**Questions:**

See weaknesses.

---

### Official Review · Reviewer_K145 · 2025-11-01

**Soundness:** 3
**Presentation:** 3
**Contribution:** 2
**Rating:** 4
**Confidence:** 3

**Summary:**

The authors in this paper propose decision language model (DLM), a framework that formulates decision-making as a dialogue-style sequence prediction problem. They perform SFT using dialogue-style datasets to enable centralized training with inter-agent context followed by a GRPO phase that trains DLM-SFT to improve robustness to OOD actions

**Strengths:**

1) Using LLMs for MARL is an interesting proposition. Reasoning models can be leveraged in complex MARL tasks for potential improvement in performance.
2) Authors achieve competitive performance on SMAC.

**Weaknesses:**

1) The generalizability of the method is a concern. How can we create prompts for complex environments where all the information is not readily available to create prompts?
2) While the method works on SMAC it should be demonstrated on diverse MARL benchmarks. Also, the method is compared to very limited number of baselines. More recent baselines should be compared with [1][2].



References:
[1] Deihim, Azad, Eduardo Alonso, and Dimitra Apostolopoulou. "Transformer World Model for Sample Efficient Multi-Agent Reinforcement Learning." arXiv preprint arXiv:2506.18537 (2025).

[2] Zhang, Yang, Chenjia Bai, Bin Zhao, Junchi Yan, Xiu Li, and Xuelong Li. "Decentralized transformers with centralized aggregation are sample-efficient multi-agent world models." arXiv preprint arXiv:2406.15836 (2024).

**Questions:**

1) How does the training and inference cost compare to other methods?
2) What will be an effective strategy to extend this method to complex environments where prompt creating is not as straight-forward as SMAC?

---

### Meta-Review · Area_Chair_XpUV · 2026-01-05

**Summary:**

The paper presents a scalable method for offline MARL, contrasting how their LLM-based approach is significantly more general than more traditional MARL methods. They present how they achieve this: first by a phase of SFT and then a phase of GRPO. The authors evaluate on a set of representative SMAC tasks where it displays similar performance to leading MARL methods, and shows improved performance on OOD tasks.

One of the main arguments of the authors is that their approach is significantly more general than traditional MARL methods. While this may be true, it is in no way something that can count as a contribution. When using LLMs within agentic tasks, it is in fact the default option to convert trajectories into conversations. Reviewer sC1f points to this clearly, citing works like SayCan (Ahn et al. 2022), but a long list can be drawn just from last year.

Another claim of the authors is that the method is significantly simpler and reaches similar performance. However, looking closer at the steps required to train the model, we can question this. The strong dependence on expert data, as pointed by Reviewer sC1f, is a significant drawback. In fact, the assumption of having access to such expert data brings into question whether we can categorize this as an offline MARL method, or simply behavior cloning. The reward function used with the GRPO algorithm, which fundamentally depend on the access to the optimal action, adds even more to this argument.

The experiments on zero-shot generalization are promising, however understand where this generalization stems from would be key. Additional baselines, comparing for example to transformers that were trained from scratch on the ChatSMAC dataset, would be interesting and would provide nuance.

**Reviewer Concerns:**

No concerns addressed.

**Reviewer Scores:**

Reviewer K145 : It's unclear if the reviewer would change score as the review was very short and did not present any convincing arguments.
Reviewer sC1f : It possible, but unlikely that the reviewer would have changed scores since the reviewer presented clear fundemental flaws with the paper (the dependence on expert trajectories).
Reviewer cTP3 : It is unlikely the rebuttal would have helped as the reviewer expresses stereotypical concerns (train on bigger models) or misunderstands some elements of the paper (the reward system in GRPO).
Reviewer i8oP : It is unlikely the rebuttal would help here too as the reviewer generated a long list of questions that make it nearly impossible to satisfy.

---

### Decision · Program_Chairs · 2026-01-26

Reject